# Quinolinecarboxamides Inhibit the Replication of the Bovine Viral Diarrhea Virus by Targeting a Hot Spot for the Inhibition of Pestivirus Replication in the RNA-Dependent RNA Polymerase

**DOI:** 10.3390/molecules25061283

**Published:** 2020-03-12

**Authors:** Simone Musiu, Yunierkis Perez Castillo, Alexandra Muigg, Gerhard Pürstinger, Pieter Leyssen, Mathy Froeyen, Johan Neyts, Jan Paeshuyse

**Affiliations:** 1KU Leuven University, Department of Microbiology and Immunology, Laboratory of Virology and Chemotherapy, Rega Institute for Medical Research, Leuven, Herestraat 49, B-3000 Leuven, Belgium; 2Bio-Cheminformatics Research Group and Escuela de Ciencias Físicas y Matemáticas, Universidad de Las Americas, 170150 Quito, Ecuador; 3Institut für Pharmazie, Abteilung Pharmazeutische Chemie, Universität Innsbruck, Innrain 80/82, A-6020 Innsbruck, Austria; 4KU Leuven University, Department of Pharmaceutical and Pharmacological Sciences, Laboratory of Medicinal Chemistry, Rega Institute for Medical Research, Leuven, Herestraat 49, B-3000 Leuven, Belgium; 5KU Leuven, Division Animal and Human Health Engineering, Laboratory for host pathogen interactions, Kasteelpark Arenberg 30, 3001 Leuven, Belgium

**Keywords:** bovine viral diarrhea virus, RNA-dependent RNA polymerase, substituted quinolinecarboxamide inhibitors

## Abstract

The bovine viral diarrhea virus (BVDV), a pestivirus from the family of *Flaviviridae* is ubiquitous and causes a range of clinical manifestations in livestock, mainly cattle. Two quinolinecarboxamide analogues were identified in a CPE-based screening effort, as selective inhibitors of the *in vitro* bovine viral diarrhea virus (BVDV) replication, i.e., TO505-6180/CSFCI (average EC_50_ = 0.07 µM, SD = 0.02 µM, CC_50_ > 100 µM) and TO502-2403/CSFCII (average EC_50_ = 0.2 µM, SD = 0.06 µM, CC_50_ > 100 µM). The initial antiviral activity observed for both hits against BVDV was corroborated by measuring the inhibitory effect on viral RNA synthesis and the production of infectious virus. Modification of the substituents on the quinolinecarboxamide scaffold resulted in analogues that proved about 7-fold more potent (average EC_50_ = 0.03 with a SD = 0.01 µM) and that were devoid of cellular toxicity, for the concentration range tested (SI = 3333). CSFCII resistant BVDV variants were selected and were found to carry the F224P mutation in the viral RNA-dependent RNA polymerase (RdRp), whereas CSFCI resistant BVDV carried two mutations in the same region of the RdRp, i.e., N264D and F224Y. Likewise, molecular modeling revealed that F224P/Y and N264D are located in a small cavity near the fingertip domain of the pestivirus polymerase. CSFC-resistant BVDV proved to be cross-resistant to earlier reported pestivirus inhibitors (BPIP, AG110, LZ37, and BBP) that are known to target the same region of the RdRp. CSFC analogues did not inhibit the *in vitro* activity of recombinant BVDV RdRp but inhibited the activity of BVDV replication complexes (RCs). CSFC analogues likely interact with the fingertip of the pestivirus RdRp at the same position as BPIP, AG110, LZ37, and BBP. This indicates that this region is a “hot spot” for the inhibition of pestivirus replication.

## 1. Introduction

The *Flaviviridae* family consists of three genera—the genus Flavivirus (including human pathogens such as dengue virus, West Nile virus, and yellow fever virus), the genus Hepacivirus (hepatitis C virus (HCV)), and the genus Pestivirus (including veterinary pathogens such as the bovine viral diarrhea virus (BVDV), and the classical swine fever virus (CSFV)). Pestivirus infections of domesticated livestock (e.g., cattle, pigs, and sheep) cause significant economic losses worldwide, [1,2,3].

BVDV is ubiquitous and causes a range of clinical manifestations (including abortion, respiratory problems, chronic wasting disease, immune system dysfunction, and predisposition to secondary viral and bacterial infections). BVDV-1 and -2 strains can cause acute fatal disease with mortality rates of 17–32% [4,5,6]. BVDV is also able to establish a persistent infection in fetuses [7]. When born, these persistently infected animals remain viremic throughout their lifespan and serve as continuous virus reservoirs. Persistently infected animals might also succumb to fatal mucosal disease if they are superinfected with a closely related BVDV strain. Vaccines are used in some countries in an attempt to control the pestivirus disease, with varying degrees of success [3]. Other containment strategies comprise quarantine and persistent-infected animal culling [3]. Currently, there are no approved antiviral drugs to control pestivirus infections. Such drugs might be an important tool to control BVDV on infected farms.

Classical swine fever is a highly infectious viral disease that affects domestic and wild pigs. CSFV is included in the list of diseases notifiable to the OIE (www.oie.int). CSFV is considered to cause one of the most devastating diseases for the pig industry, throughout the world, both from an economical and sanitary point of view [8]. The disease is endemic in Asia and is prevalent in many countries of Central and South America. In contrast to North America, where CSFV was eradicated several decades ago, the European Union (EU) still has an ongoing progressive eradication program that started in the early 1990s [8]. The most efficient vaccines currently available against CSFV are live attenuated vaccines [8]. However, many efforts have recently been put into the development of new and safer marker vaccines against CSFV, along with improved diagnostic tools [9,10]. Broad-spectrum pestivirus inhibitors might also be considered to control outbreaks with CSFV, in otherwise disease-free areas. Other possible uses of anti-pestivirus drugs might be (i) to treat valuable animals infected with pestiviruses in zoologic collections, (ii) to treat expensive animals in breeding programs and *in vitro* embryo production [11], to (iii) cure established cell lines from contaminating pestiviruses [12,13,14].

Several classes of pestivirus inhibitors [14] have been reported. They either target a cellular protein/enzyme, i.e., α-glycosidase (which is involved in the maturation of virions [15]), as well as viral encoded enzymes such as the NS3 protease and helicase/NTPase [16], or the NS5B RNA-dependent RNA polymerase (RdRp). Polymerase inhibitors include nucleoside [14] and non-nucleoside inhibitors, such as *N*-propyl-*N*-[2-(2*H*-1,2,4-triazino[5,6-*b*]indol-3-ylthio)ethyl]-1-propanamine (VP32947) [17], a thiazole urea derivative [18], a cyclic urea derivative [19], imidazo-pyridines (BPIP) [20], ethyl 2-methylimidazo[1,2-*a*]pyrrolo[2,3-*c*]pyridin-8-carboxylate (AG110) [21], pyrazolotriazolopyrimidinamine (LZ37) [22], 2-(2-benzimidazolyl)-5-[4-(2-imidazolino)phenyl]furan (DB772) [23], 5,6-dimethoxy-1-indanone [24,25], 2-phenylbenzimidazole [26], substituted 2,6-bis(benzimidazol-2-yl)pyridines [27], benzimidazole derivative [28], and arylazoenamine derivatives [29].

BVDV strains that are resistant to the majority of these non-nucleosidic RdRp inhibitors all carry mutations in the fingertip domain of the viral RdRp. However, most of these inhibitors do not inhibit the *in vitro* activity of the recombinant viral polymerase but are able to inhibit the activity of the BVDV replicase complex (RC), in a dose dependent manner [20,21,22,27,30]. The fingertip domain of the polymerase is crucial for the function of the polymerase and the viral RC. This domain is, thus, apparently a “hot spot” binding site for selective inhibitors of pestivirus replication [20,21,22,27,30].

Here, we report on the antiviral characteristics and mode of action of a series of quinolinecarboxamide analogues as a new class of chemicals that inhibit the replication of pestiviruses.

## 2. Materials and Methods

### 2.1. Compounds

2-(2-furanyl)-*N*-(6-methyl-2-benzothiazolyl)-4-Quinolinecarboxamide (T0505-6180/CSFCI) and 2-phenyl-*N*-(1-phenyl-1*H*-pyrazol-3-yl)-4-Quinolinecarboxamide (TO502-2403/CSFCII) were purchased from Ambinter (Orléans, FRANCE). The structural formula of both compounds is depicted in Figure 1. Details on the synthesis of the quinolinecarboxamide analogues can be obtained from Gerhard Puerstinger (University of Innsbruck, Innsbruck, Austria). Structural information for each analogue synthesized can be found in Appendix A. The synthesis of LZ37 [7-[3-(1,3-benzodioxol-5-yl)propyl]-2-(2-furyl)-7*H*-pyrazolo[4,3-*e*][1,2,4]triazolo[1,5-*c*]pyrimidin-5-amine] [22], BPIP (5-[(4-bromophenyl)methyl]-2-phenyl-5*H*-imidazo[4,5-*c*]pyridine [20], and ethyl 2-methylimidazo[1,2-*a*]pyrrolo[2,3-*c*]pyridin-8-carboxylate (AG110) [21], were reported earlier. 3′-deoxyguanosine-5’-triphosphate (3’-dGTP) and 2’-*C*-methylguanosine-5’-triphosphate (2′-*C*-me-GTP) were purchased from Trilink (San Diego, CA).

### 2.2. Cells and Viruses

Madin–Darby bovine kidney (MDBK) cells were grown in a minimal essential medium (MEM), supplemented with 5% heat-inactivated fetal bovine serum (FBS) (Integro, Zaandam, The Netherlands). The BVDV National Animal Diseases Laboratory (NADL) strain was obtained from the Veterinary and Agrochemical Research Center (Sciensano, Ukkel, Belgium). BPIP-resistant BVDV (BPIP^res^) was produced from a pNADLp15a plasmid containing the F224S mutation in the NS5B gene, as described previously [20]. Porcine kidney cell line (PK15), in conjunction with the CSF reference-strain Alfort_187_ (subgroup 1.1) were cultured, as described before [31].

### 2.3. Antiviral Assay 

The experiments were performed, as previously described [20,21,22,27,30]. In brief, MDBK cells were seeded at a density of 5 × 10^3^ per well in 96-well cell culture plates. Following 24 h incubation, at 37 °C and 5% CO_2_, the medium was removed and three-fold serial dilutions of the test compounds were added in a total volume of 100 µL, after which the cells were infected with the BVDV NADL virus (100 cell culture infectious dose 50%, CCID_50_). After 3 days, the medium was removed and the cytopathic effect (CPE) was quantified using the MTS/PMS method (Promega, Leiden, The Netherlands). The 50% effective concentration (EC_50_) was defined as the concentration of compound that offered 50% protection of the cells against virus-induced CPE and was calculated using linear interpolation.

### 2.4. Cytotoxicity Assay 

Assays were carried out, as described previously [20,21,22,27,30]. In brief, MDBK cells were seeded at a density of 5 × 10^3^ per well in 96-well cell culture plates, in MEM containing 5% FBS. 24 h later, serial dilutions of the test compounds were added. The cells were allowed to proliferate for 5 days at 37 °C, after which the overall metabolic activity of the cells was quantified by means of the MTS/PMS method (Promega, Leiden, The Netherlands). The 50% cytotoxic concentration (CC_50_) was defined as the concentration of compound that inhibited the proliferation of exponentially growing cells by 50%, and was calculated using linear interpolation.

### 2.5. 3D QSAR Model

Otherwise noted, default parameters provided with all software were used for modeling. One initial 3D structure per compound was obtained with OpenEye’s Omega [32]. Conformational exploration and alignment of the compounds was performed with the Open3DALIGN software [33]. A quenched molecular dynamics (QMD) conformational search was carried out for each compound using TINKER, with an implicit solvent model [34]. A total of 200 QMD simulations of 100 ps length were run per compound. The MIXED method was used for structural alignment with Open3DALIGN.

Next, compounds with undetermined IC_50_ (>100 μM or <0.02 μM) were removed from the modeling process and bioactivities were transformed to pIC_50_ (
${pIC}_{50}=-\log_{10}{IC}_{50}$
). Given the negative effect of the activity cliffs in the QSAR modeling, activity cliff generators (ACGs) were removed from the dataset [35,36]. ACGs were defined as compounds sharing a similarity greater than 65% with any other in the dataset and with pIC_50_ differences greater than two in between them. Similarities were computed from the 1024 bits Extended Connectivity Fingerprints, using MayaChemTools [37]. This step was accomplished by employing a home-developed software that searches for ACGs and removed them sequentially (one at a time). In each cycle, the program removed the compound, forming the largest number of cliffs with the rest of the dataset. If more than one ACG shared the same maximum number of formed cliffs, the one with the lowest potency was removed. In case more than one ACG shared the same minimum activity and generated the same number of cliffs, one of them was randomly removed. The list of ACGs is updated at each iteration and the process is repeated until no ACG is present in the dataset.

The compounds remaining after the removal of those with undetermined activity values and activity cliff generators were used for 3D-QSAR modeling. These models were generated with the Open3DQSAR suite [38], which performs partial least squares (PLS) regression models from molecular interaction fields (MIFs). The best alignment produced by the Open3DALIGN in the previous step was used for the 3D-QSAR model generation. Prior to modeling, the dataset was randomly split into training and external validation sets.

Grid step was set to 1 Å and the grid box exceeded the largest molecule by 5 Å. Steric and electrostatic molecular mechanics MIFs were computed using the Merck force field (MMF94) and the probe for MIFs computation was set to an alkyl carbon (charge + 1). Variables were filtered to discard those with Van der Waals energies above 10^4^ kcal/mol and the corresponding points on the electrostatic MIF were also removed from the analyses. Variables spanning up to four levels and those with a standard deviation lower than 0.1 were discarded. Energies greater than 30 kcal/mol or lower than –30 kcal/mol were set to 30 kcal/mol and –30 kcal/mol, respectively, in both MIFs. Additionally, values of energy between –0.05 and 0.05 kcal/mol were set to 0 in the two MIFs. As a final filter, the variables were scaled by employing the Block Unscaled Weighting (BUW) option of the Open3DQSAR.

The initial PLS models considered 10 principal components and were trained using the training set compounds. The optimal number of principal components (PCs) was determined based on the value of the leave one out cross-validation R^2^ (q^2^_LOO_). Variables were then grouped using the smart region definition procedure implemented in the Open3DQSAR, by taking into account the previously identified optimal number of PCs. The grouped variables were subject to a selection procedure, according to the fractional factorial design using leave many out (LMO) cross-validation with 20 runs and 5 groups. Only the selected variables in the previous step were kept in the dataset. Afterwards, the PLS model was re-computed for the optimal number of PCs and the external validation set was predicted.

### 2.6. Isolation of TO502-2403 ^res^ /CSFCII ^res^ TO505-6180 ^res^ /CSFCI^res^ BVDV 

CSFC-resistant (CSFC^res^) virus was selected, as previously described [20,21,22,27,30]. In brief, CSFC^res^ BVDV was generated by culturing wild-type BVDV in MDBK cells, in the presence of increasing concentrations of the compound in a 48-well plate. After 3 days of cultivation, cultures were freeze-thawed. Lysates of the infected and treated cultures that exhibited a cytopathic effect under drug pressure were used to infect new cell monolayers. These were further incubated in the presence of increasing concentrations of the compound. The procedure was repeated for 25 consecutive passages, until a drug-resistant virus was obtained. BVDV strains resistant to other inhibitors were passaged in the presence of the corresponding inhibitor and sequenced in parallel to the newly selected CSFC-resistant variants.

### 2.7. RNA Isolation

Viral RNA was isolated from the cell culture supernatant, using a QIAamp viral RNA minikit (Qiagen, Venlo, The Netherlands). Total cellular RNA was isolated from the cells, using an RNeasy minikit (Qiagen, Venlo, The Netherlands), according to the manufacturer’s instructions. To extract CSFV RNA, the total cellular RNA content was isolated from the infected cell cultures, using the RNeasy Mini kit (QIAgen, Venlo, The Netherlands), according to the manufacturer’s instructions.

### 2.8. RT-qPCR

RT-quantitative PCR (RT-qPCR) was performed, as previously described [20,21,22,27,30]. In brief, a 25-µL RT-qPCR reaction was composed of a 12.5 µL 2 × reaction buffer (Eurogentec, Seraing, Belgium), 6.3 µL H_2_O, 300 nmol/L forward and reverse primer, 300 nmol/L TaqMan probe and 5 µL total cellular or viral RNA extract. The RT step was performed at 48 °C for 30 min, denatured for 15 min at 95 °C and subsequent PCR amplification of 40 cycles of denaturation was done at 94 °C for 20 s, and annealing and extension was done at 60 °C for 1 min in an ABI 7500 FAST Real-Time PCR System. The EC_50_ was defined as the concentration of compound that reduced the amount of viral RNA by 50%, as compared to an infected untreated control and was calculated using linear interpolation.

### 2.9. Sequencing

PCR fragments that cover the entire nonstructural protein coding region of the BVDV genome were generated and analyzed, using the cycle sequencing method (ABI Prism BigDye Terminator Cycle Sequencing Ready Reaction Kit). Both DNA strands were sequenced. Sequence data were obtained using an ABI 373 Automated Sequence Analyzer (Applied Biosystems), and sequences were analyzed using the Vector NTI software package (Invitrogen, Merelbeke, Belgium).

### 2.10. RNA-Dependent RNA Polymerase Assay

BVDV (NADL) RNA-dependent RNA polymerase (RdRp) was expressed and purified, as described before [39]. These experiments were performed, as previously described [20,21,22,27,30]. In brief, the purified BVDV polymerase (100 nM) was mixed with 100 µM GTP (containing 8.3 µM of [^3^H]GTP, Perkin Elmer) and increasing concentration of the inhibitor (0.2 µM, 0.7 µM, 2 µM, 6 µM, 19 µM, 56 µM, 165 µM or 500 µM) in 50 mM Hepes pH 8.0, 10 mM KCl, 10 mM DTT, 1 mM MgCl_2_, 2mM MnCl_2_, and 0.5% igepal (Sigma). The enzyme mix and inhibitors were pre-incubated, in order to favor enzyme-inhibitor interactions before RNA binding. Reactions were started by the addition of 100 nM of poly(C) (about 500 nt in size) template. Reactions were incubated at 30 °C and stopped by addition of 50 mM EDTA. Samples were transferred onto DE-81 filters, washed with 0.3 M ammonium formate solution, and dried. Radioactivity bound to the filter was determined by liquid scintillation counting.

### 2.11. Replication Complex Assay

The replication complex (RC) assay is similar to the one described before [20,22]. In brief, BVDV-infected MDBK cells were suspended in an ice-cold hypotonic buffer A (10 mM Tris, 10 mM NaCl pH 7.8) and were further disrupted by 50 strokes with a Dounce homogenizer. The disrupted cells were pelleted by centrifugation at 1,000× *g* for 5 min at 4 °C. The supernatant fraction, containing cytoplasmic material and plasma membranes, was concentrated by high-speed centrifugation at 200,000× *g* for 20 min at 4 °C. The pellet was resuspended in 120 µL of hypotonic buffer B (10 mM Tris, 10 mM NaCl pH 7.8 and 15% glycerol) and used for an RNA polymerase assay. Replicase reactions were carried out, as described before, with some modifications [20,22]. The [α-^33^P]UTP (3,000 mCi/mmol) was replaced by [α-^33^P]CTP (3,000 mCi/mmol). The following incubation, RNA extraction, RNA analysis, and quantification were performed, as previously described [20,22].

### 2.12. Molecular Modeling

The published X-ray structure of the BVDV RdRp [PDB entry 1S48 [40]] was used for the docking experiments. Selenium atoms in the selenomethionine residues were modified back to sulfur atoms, to get methionine residues. Target structure and ligand BBP was prepared for docking by Autodocktools (Gasteiger charges were added, atom types assigned for use with autodock 4.2). The docking site for BBP was defined as a cube centered at Phe224A.CD1, dimensions 60 × 60 × 60, spacing 0.375 Å [41]. Flexible dockings of CSFC analogues with 2 active torsion angles were performed by using Autodock 4.2. Several runs (100) using a Lamarckian genetic algorithm were performed and the best docking (lowest energy score) was selected.

## 3. Results 

### 3.1. In Vitro Antiviral Activity of Quinolinecarboxamides

Previously, we performed a large-scale, CPE-based antiviral screen [21], during which we identified two quinolinecarboxamides [TO502-2403 (CSFCII)/ TO505-6180 (CSFCI), Figure 1] that resulted in selective *in vitro* inhibition of BVDV-1 replication. Both analogues inhibited the replication of the BVDV-1 (strain NADL) in a MDBK cell, in a dose-dependent manner. The EC_50_ for TO502-2403 (CSFCII) was 0.2 µM (SD 0.06 µM) (Figure 2A) and 0.07 µM (SD 0.02 µM) for TO505-6180 (CSFCI, Figure 2A). Both compounds were devoid of cytotoxicity (CC_50_ > 100 µM). The anti-BVDV activity of both molecules was further corroborated by measuring the effect of the compounds on infectious virus yield and on viral RNA synthesis. The EC_50_ for inhibition of infectious virus yield in the culture supernatant was 0.2 µM (SD 0.09 µM) for TO502-2403 (CSFCII) (Figure 2B) and 0.6 µM (SD 0.1 µM) for TO505-6180 (CSFCI, Figure 2B). The EC_50_ for inhibition of viral RNA formation in cells and in the culture supernatant for TO502-2403 (CSFCII) was 0.06 µM (SD 0.01 µM) and 0.05 µM (SD 0.02 µM), respectively, showing a very similar pattern of inhibition (Figure 2C,D). The EC_50_ for the inhibition of intra- and extracellular viral RNA formation for TO505-6180 (CSFCI, Figure 2C,D) was 0.03 µM (SD 0.01 µM) and 0.06 µM (SD 0.005 µM), respectively. 

Both quinolinecarboxamide analogues modestly inhibited the *in vitro* replication of the classical swine fever virus (CSFV) and proved inactive *in vitro*, against the hepatitis C virus that belongs to BVDV, and CSFV that belongs to the family of *Flaviviridae*. Furthermore, quinolinecarboxamide analogues proved inactive against a panel of unrelated RNA and DNA viruses (data not shown).

### 3.2. 3D-QSAR Analysis of All Tested Analogues of CSFC Lead Molecules

To better understand the relationships between the structural properties of the CSFC analogues tested and their biological activity, a 3D Quantitative structure-activity relationships (QSAR) model was build, following previously described methodology. From the 145 tested compounds, 23 were removed from the modeling process due to undetermined EC_50_ values. In addition, 17 more compounds were removed to eliminate the activity cliffs generators from the dataset. The final model contained four principal components and showed a good statistical performance with R^2^_Train_ = 0.82, q^2^_LOO_ = 0.66, and q^2^_Ext_ = 0.73. The detailed dataset composition and partitioning, including the compounds removed due to any of the reasons listed above, as well as model predictions are provided as the Appendix A.

According to the 3D-QSAR model, the steric effect contributed to 72% of bioactivity, while the contribution of the electrostatic effect was 28%. This model is summarized in Figure 3 for the CSFCII and CSFCI derivatives. In both cases, bulky substituents at the R_1_-position, positively influenced the bioactivity, and their presence could be considered critical for the anti-BVDV bioactivity. This effect could be seen from the comparison of compounds **43**, which lacked a substituent at R_1_, with compounds **85** and **86**. As a result of the surrounding electrostatic field of position R_1_, negative charged moieties in the para position of aromatic rings used in R_1_ could increase activity.

The positioning of an electronegative molecular interaction field (MIF) close to R_1_, suggests that non-phenyl aromatic substituents at R_1_, such as furan or pyridine with electronic clouds displaced toward the N or O atoms could increase bioactivity (see compounds **3**, **36**, **110** and **132**).

Substituents at R_3_ could have a negative impact on the anti-BVDV bioactivity, due to the presence of a steric unfavorable region close to this position. Small substituents are well tolerated, however, larger atoms such as I (compound **22** with a larger halogen compared to other analogues that carry a halogen substituent at this position) decreased activity. A bulky substituent at position R_3_ had a major negative impact on bioactivity (compound **98**). Additionally, a steric favorable region was observed around R_4_. Small substituents at this position improved the bioactivity of the compounds (compound **20**, **24** and **25**).

For the CSFCII analogues, bulky substituent at position R_2_ also had a positive effect on the bioactivity (critical for bioactivity). For example, the lack of such a moiety at R_2_ in analogues 44 and 47, compared to compounds **1**–**9** abolished the bioactivity of these compounds. However, too large substituents like those present in compounds **57** (6-CF3) and **59** (6-CH3O2S) had a steric unfavorable effect, as suggested by the existence of a delimited steric favorable region near position R_2_. A para-negative substituent in R_2_ could have a negative effect on bioactivity, given that it occupied a region with positive electrostatic MIF.

Several positive and negative electrostatic fields perpendicular to each other were observed around the R_1_ and R_2_ position. These fields were orientated as such that they favored stacking of the pi-electrons from the aromatic rings between them. For this reason, substituents with aromatic properties could be preferable in position R_1_ and R_2_.

In addition to the previously discussed influence of substitutions at R_1_, R_3_, and R_4_, for the CSFCI derivatives, small non-bulky substitutions at any position—R_21_, R_22_, R_23_ and R_24_—contributed to improved bioactivity. In contrast, bulky substitutions at the later positions highly decreased bioactivity. This effect could be observed from the comparison of compounds **51**, **57**, **59** and **112**. According to the steric MIF surrounding positions R_21_, R_22_, R_23_, and R_24_, the simultaneous substitution of these positions in the same compound could increase bioactivity. Additionally, electronegative substitutions were favored at position R_24_ and disfavored at R_23_. Finally, the presence of the pyrazol moiety in the CSFCII derivatives and of the benzothiazole group in the CSFCI analogues attached to the quinoline-4-carboxamide scaffold were essential for bioactivity.

### 3.3. Isolation and Characterization of Drug-Resistant Viruses

To decipher the mechanism through which the CSFC analogues inhibited viral replication, escape mutants to the drugs were selected by propagating BVDV (strain NADL) for 25 passages, in increasing concentrations of the drug (from 1.8 to 30 µM) and were characterized. The TO502-2403/CSFCII drug-resistant virus variants (TO502-2403^res^) proved more than 500-fold less susceptible to the inhibitory effect of TO502-2403/CSFCII, than the parent wild-type strain and about 1580-fold less susceptible to BPIP, 11-fold less susceptible to LZ37, >63-fold less susceptible to AG110, 200-fold less susceptible to BBP, and >1429-fold less susceptible to TO505-6180/CSFCI (Table 1). The TO505-6180/CSFCI drug resistant variants (TO505-6180 ^res^) were >1429-fold less susceptible to the inhibitory effect to TO505-6180/CSFCI, compared to the parent wild-type strain and about 900-fold less susceptible to BPIP, 11-fold less susceptible to LZ37, 16-fold less susceptible to AG110, >333-fold less susceptible to BBP, and 160-fold less susceptible to TO502-2403/CSFCII (Table 1). Previously selected variants that were resistant to different inhibitors showed cross-resistance to TO505-6180/CSFCI and to TO502-2403/CSFCII, ranging from 19-fold (i.e., TO502-2403/CSFCII against AG110^res^) to >1429-fold (Table 1). To identify the molecular changes that are responsible for the drug-resistant phenotype, we compared the TO502-2403^res^ and TO505-6180^res^ genome sequences to the parental wild-type strain (GenBank accession no. AJ781045). For TO502-2403^res^, we identified a transition of T to the C mutation, at position 10,862, which resulted in an amino acid change of phenylalanine (F) to proline (P) at amino acid residue 224 in the NS5B gene.

For TO505-6180^res^, two mutations in the NS5B polymerase gene were identified, one at the position 10,862 [T to A substitution (F224Y)] and the other at position 10,982 [A to G substitution (N264D)]. Interestingly, the mutation at position F224 was also identified in the genome of the BPIP^res^ [21] and the LZ37^res^ virus [23].

### 3.4. Effect of the CSFC Analogues on the BVDV RdRp and Replication Complexes

As the mutations identified in the drug-resistant virus variants were all located in the NS5B gene, that encoded the viral RdRp, we next studied the inhibitory effect of both CSFC analogues on the *in vitro* polymerase activity of the enzyme. TO502-2403/CSFCII, TO505-6180/CSFCI, and the nucleotide analogue 3′-dGTP (which was included as a positive control) were tested for potential inhibitory activity against the highly purified BVDV RdRp, by using poly(C) as a template. The 50% inhibitory concentrations for the BVDV polymerase activity were < 1 µM for 3′-dGTP. TO502-2403/CSFCII and TO505-6180/CSFCI had no effect on the activity of the viral polymerase (Figure 4). 

Since TO502-2403/CSFCII and TO505-6180/CSFCI did not inhibit the activity of the purified BVDV RdRp, we tested the effect of the compound on viral RCs isolated from MDBK cells that had been infected with the wild-type virus or with the selected TO502-2403^res^ or TO505-6180^res^ virus BVDV strain. TO502-2403/CSFCII inhibited the activity of the BVDV WT replication complexes in a dose-dependent manner, with a maximum inhibition ~90% at 10 µM (Figure 5A,B). In contrast, the activity of RCs isolated from MDBK cells that had been infected with the TO502-2403^res^ strain were not susceptible to the inhibitory effect of TO502-2403/CSFCII (Figure 5A,B). TO505-6180 inhibited the activity of the BVDV wt RCs in a dose-dependent manner (Figure 5C,D), whereas RCs isolated from MDBK cells that had been infected with the TO505-6180^res^ virus was 3 to 4-fold less susceptible to the inhibitory effect of TO505-6180, as compared to BVDV wt RCs (Figure 5C,D).

### 3.5. Computational Docking of CSFC Analogues in the BVDV RdRp Crystal Structure

Based on the crystal structure of the BVDV RdRp (PDB 1S48) [24,40,42], the amino acid position F/P/Y224 was located in a small cavity near the fingertip domain of the BVDV polymerase. The F224 position was already shown to be implicated in resistance of BVDV to VP32947, BPIP, and LZ37 [17,20,22]. Position N264 was located in the conserved motif I of the finger domain of the BVDV polymerase.

Docking of TO502-2403/CSFCII in this cavity revealed the following possible interactions between the polymerase and the compound: *(i)* a hydrogen bond between the N in the middle of the scaffold of TO502-2403/CSFCII and the residue A392 via the main chain O and *(ii)* a stacking interaction between the aromatic ring of the F224 and the aromatic ring of the quinoline-4-carboxylic acid scaffold (Figure 6). When the phenylalanine mutated into a proline the stacking interaction would be lost, thereby, also destabilizing the hydrogen bond with A392. Hence, when the F224P substitution occurred, the interaction between the TO502-2403/CSFCII inhibitor and the RdRp would no longer be possible

Docking studies of TO505-6180/CSFCI in the same cavity of the BVDV polymerase, as described above, resulted in a model that revealed the following possible interactions between the polymerase and the inhibitor. Stacking interactions with F224 were observed, together with a hydrogen bond from the O in the furan ring of the inhibitor and the amide N of Asn264. In such cases where one or both of these interactions were lost due to the mutation N264D or F224Y, binding of this inhibitor was no longer favourable. Here, we speculated that the presence of the OH group in the side chain of the mutated Y224 might interact with the surrounding protein matrix, for instance forming an H-bond with the A392 backbone CO group and forcing the Y224 into another rotameric state, where the stacking interaction with the inhibitor is not possible.

## 4. Discussion 

During the course of a large screening effort dedicated to identifying pestivirus inhibitors, two quinolinecarboxamide analogues (i.e., TO502-2403/CSFCII and TO505-6180/CSFCI) were identified as selective *in vitro* inhibitors of the replication of pestiviruses. Both independently identified hit compounds inhibited the *in vitro* BVDV-1 and CSFV replication but proved inactive against related viruses (the HCV) and a panel of unrelated RNA and DNA viruses. Based on the hit molecule TO502-2403/CSFCII, a series of 104 new analogues were synthesized in an attempt to optimize this compound class for inhibitory properties and the selectivity on the replication of BVDV. A 3D-QSAR model, Figure 3, was derived from the antiviral activity of these analogues evaluated against BVDV. From this model, we could extrapolate that the bio-activity of the lead molecule could be improved by introducing bulky aromatic groups with negatively charged substituents in para in R_1_, and by adding bulky aromatic groups that lacked a negatively charged, small substituent in para at R_2_ of the CSFCII analogues. Position R_3_ could only tolerate small substituents, while small substituents in position R_4_ favored bioactivity. In the case of the CSFCI derivatives, only small non-bulky substitutions were allowed at the R_21_, R_22_, R_23_, and R_24_ positions.

Parallel drug-resistant BVDV variants for both hit molecules were selected and a geno- and phenotypic analysis was performed. *In vitro* selected TO502-2403^res^ carried an amino acid transition from phenylalanine (F) in the WT to proline (P), at position 224, the TO505-6180^res^ virus carried a phenylalanine (F) to tyrosine (Y) mutation at position 224 (F224Y), and an asparagine (N) to aspartic acid (D) mutation at position 264 (N264D) of the viral RNA-dependent RNA polymerase. Interestingly, amino acid position F224 was previously shown to be involved in the antiviral resistance of BVDV to VP32947 [17], BPIP [20], and LZ37 [22], whereas mutation N264D was reported to be key in the phenotypic resistance against DB772 [23], 5,6-dimethoxy-1-indanone [26], and 2-phenylbenzimidazole [28]. Here, for the first time we reported the selection of a drug-resistant virus that combines both the F224P/Y and the N264D mutation. Both TO502-2403^res^ and TO505-6180^res^ were resistant to inhibition by other compounds (i.e., LZ37, BPIP, AG110, and BBP) that target the same domain within the RdRp. However, TO505-6180^res^ proved relatively more susceptible to inhibition by AG110 than TO502-2403^res^ (16-fold vs. 63-fold). Likewise, BVDV variants resistant to various inhibitors targeting the same hotspot in the RdRp were to a much lesser extent inhibited by both CSFC analogues tested, as compared to WT BVDV. Remarkably, AG110^res^ and BBP^res^ were relatively more susceptible to inhibition by TO502-2403. An explanation for the peculiar (cross)-resistance pattern observed (Table 1), regardless of the chemical structure of these inhibitors, could be that three common interaction sites constituted the pocket that binds all inhibitors described above. The location of the first interaction site of the pocket is based on the AA position F224, which is located in the fingertip region of the RdRp. The fingertip region interacts with the thumb domain, with which it encloses the active site of the RdRp, forming an entrance to the template-binding channel. The fingertip region is presumptive to be engaged in the flexibility of the finger domain responsible for template/product translocation [42], protein–protein interactions, or dimerization of the RdRp in the replication complex [40,41,43,44], enabling the assembly of an active replication complex. Mutation (F224P/Y) present in the resistance virus variants selected under antiviral pressure with CSFC-analogues, thus, might cause conformational changes in the fingertip region of RdRp, disrupting one of the above-mentioned functions. 

Furthermore, the F224 position in the RdRp was located at a distance of only 14 Ǻ from E291 (which is mutated in strains resistant to AG110) which was located in the conserved motif II of the RdRp. A second site of interaction, based on the location of mutation I261M [27,45], P262A [45], and N264D [45], were located in motif I of the RdRp, close to the NTP *i* + 1 binding site, indicating a tentative role in binding the incoming NTP [40]. The third interaction site was based on the E291G mutation reported for AG110^res^ [21], which was part of the RdRp motif II that, through contacts with the phosphate backbone and bases, might have been involved in template binding [40]. Thus, the fact that the E291G mutation (identified in AG110^res^ viruses) was located in motif II of the RdRp and given the importance of the interaction of AG110 with residue(s) from motif II, might explain the relative increased susceptibility of TO505-6180^res^ to AG110 and that of AG110^res^ and BBP^res^ to TO502-2403. Especially since AG110^res^ combines a mutation in motif II with a different mutation on the fingertip of the polymerase and the BBP^res^ has no mutations in the fingertip region of RdRp, but only in motif I. Given the flexibility of the finger domain of the RdRp, it might as well be that mutations in one binding site of the pocket influences the conformation of the entire pocket, possibly mediated through additional interactions with the RNA template.

For TO502-2403^res^, a single mutation in this region of the RdRp proved sufficient to confer complete resistance.

Even if this mutation provided strong evidence that the viral polymerase was the target of interaction with the class of quinolinecarboxamides, similar to the reference compound BPIP [20], AG110 [21], and LZ37 [22], none of these compounds had any inhibitory effect on the highly purified BVDV RdRp. 

Since, the quinolinecarboxamides did not inhibit the activity of the purified BVDV RdRp in in-vitro enzymatic assays, we tested the effect of the compound on viral RCs. Indeed, within the intact cell, the RdRp functions in the context of membrane-bound RCs, which consist of several virus proteins, host proteins, and various forms of viral RNA. Both CSFC analogues inhibited the function of the wild-type RC but not that of the mutant strain. The observation that quinolinecarboxamides inhibited the function of RC but not that of the purified RdRp, might be explained by the fact that, following binding to the NS5B, the compound might disrupt the protein–protein interactions and the stability between the several virus-encoded proteins, and might interfere with the function and the proper formation of a functional replication complex. Another possible explanation might be that the binding of quinolinecarboxamides to the polymerase, results in reduced finger flexibility or impairment of the ability of the polymerase to translocate its template/product during polymerization.

In conclusion, a class of quinolinecarboxamides that are selective inhibitors of the replication of pestiviruses was identified. Quinolinecarboxamides were cross-resistant with a number of earlier reported pestivirus replication inhibitors, such as BPIP, AG110, BBP, and LZ37. All these inhibitors were selected for mutations within a region of the finger domain of the RdRp, which was only 7 Å across. Quinolinecarboxamides can interact with the fingertip of the BVDV-RdRp region, confirming that this cavity is a “hot spot” for inhibition of pestivirus replication.

## Figures and Tables

**Figure 1 molecules-25-01283-f001:**
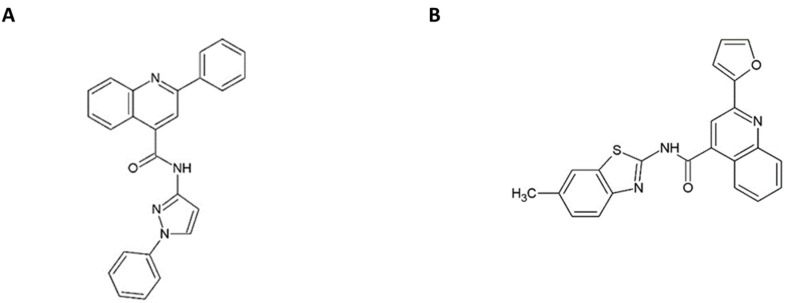
Structural formula of the quinolinecarboxamide A TO502-2403 (CSFCII) B TO505-6180 (CSFCI).

**Figure 2 molecules-25-01283-f002:**
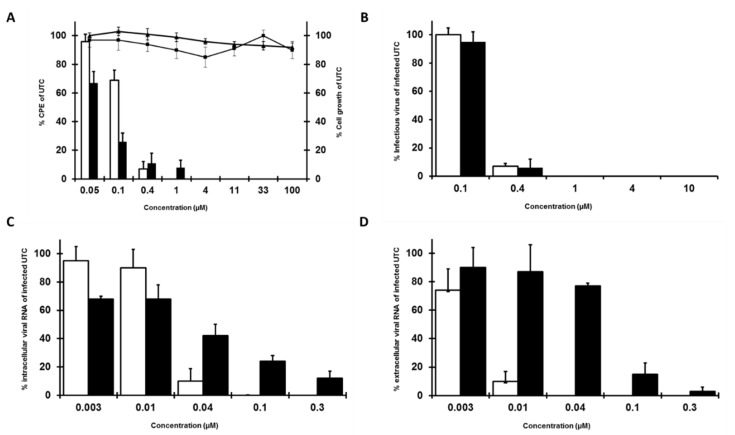
Effect of TO502-2403 (CSFCII, open bars, diamonds) or TO505-6180 (CSFCI, black bars, squares) on (**A**) BVDV (NADL)-induced cytopathic effect (CPE) formation in the MDBK cells (bars) and on the proliferation of exponentially growing Madin–Darby (MDBK) cells TO502-2403 (CSFCII, diamonds) or TO505-6180 (CSFCI, squares), (**B**) on infectious virus yield, (**C**) on intracellular viral RNA, and (**D**) on the release of extracellular viral RNA.

**Figure 3 molecules-25-01283-f003:**
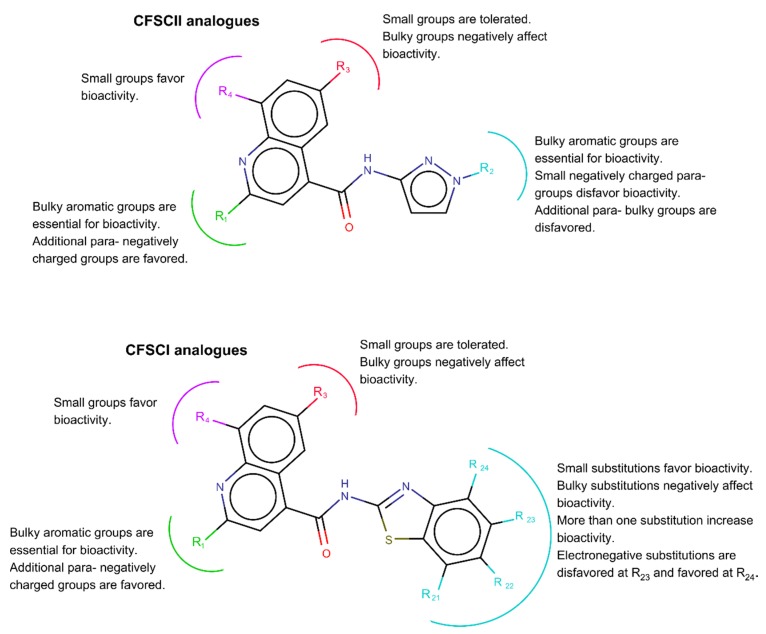
Summary of the 3D-QSAR analyses performed on the CSFCII (**top**) and CSFCI (**bottom**) analogues tested.

**Figure 4 molecules-25-01283-f004:**
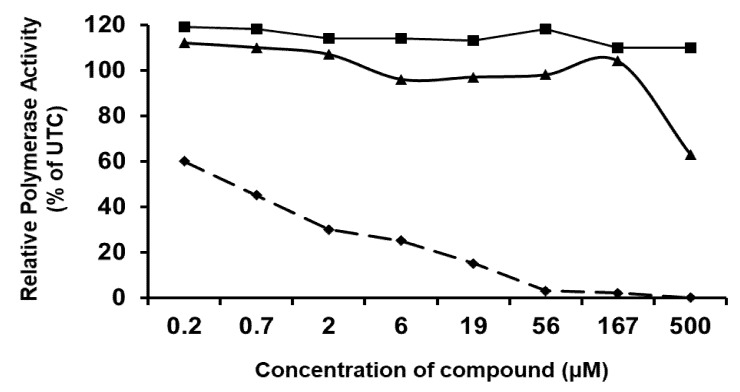
A. Effect of TO502-2403 (CSFCII) (filled triangles), TO505-6180 (CSFCI) (filled squares), and 3′-dGTP (filled diamonds) on the activity of the purified BVDV RdRp, using poly(C) as a template. Data are from a typical experiment are expressed as a percentage of the untreated control.

**Figure 5 molecules-25-01283-f005:**
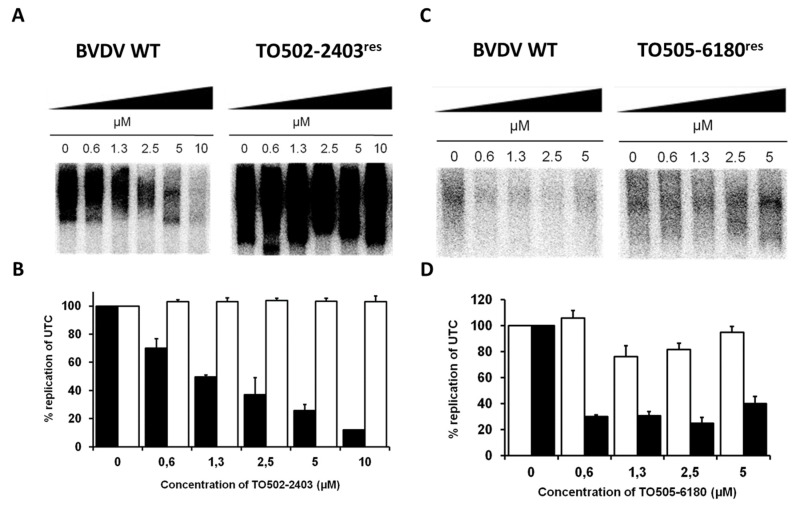
(**A**) Effect of TO502-2403 (CSFCII) on the activity of RCs isolated at 14-h post-infection from MDBK cells that had been infected with wild-type BVDV (NADL) or with the selected TO502-2403^res^ BVDV strain. Reaction product of the RC assay was separated on a 1% agarose-glyoxal denaturing gel. RCs for these assays were either isolated from the MDBK cells infected with the wild-type virus or with the TO502-2403^res^ BVDV strain. (**B**) Densiometric analysis of the autoradiograph depicted in panel A and also from a second assay (picture not shown). Black bars represent the activity of the wild-type BVDV (NADL) RCs, and open bars represent the activity of RCs from the cell infected with TO502-2403^res^ virus. (**C**) Effect of TO505-6180 (CSFCI) on the activity of RCs isolated at 14-h post-infection from the MDBK cells that had been infected with wild-type BVDV (NADL) or with the selected TO505-6180^res^ BVDV strain. (**D**) Densiometric analysis of the autoradiograph depicted in panel C and also from a second assay (picture not shown). Black bars represent the activity of the wild-type BVDV (NADL) RCs, and open bars represent the activity of the RCs from the cell infected with the TO505-6180^res^ virus. Data are mean values ± SD of two independent experiments. UTC—untreated control; WT—wild-type.

**Figure 6 molecules-25-01283-f006:**
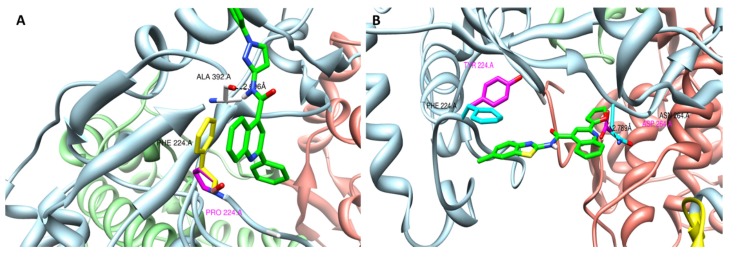
Modelling of (**A**) TO502-2403 (CSFCII) near the position F224P in the RNA-dependent RNA polymerase and (**B**) TO505-6180 (CSFCI) near the position F224Y and N264D in the RNA-dependent RNA polymerase.The different domains of the BVDV polymerase are colored as follows—the N-terminal domain is in yellow (residues 92 to 138), the finger domain is in blue (residues 139 to 313 and residues 351 to 410), the palm domain is in green (residues 314 to 350 and residues 411 to 500), and the thumb domain is in red (thumb, residues 501 to 679). The ligand carbon atoms are coloured green. The hydrogen bond is indicated with the dotted line.

**Table 1 molecules-25-01283-t001:** Susceptibility of the bovine viral diarrhea virus (BVDV) wild-type (NADL) and drug-resistant BVDV strains to different inhibitors.

Virus Strain and Mutations	EC_50_ (µM) of the Indicated Inhibitor ^a^						
	LZ37	BPIP	AG110	BBP	TO502-2403	TO505-6180	2′-*C*-Met-Cyt
**LZ37^res^** F224P					60 (9) 303x	>100 >1429x	
**BPIP^res^** F224S I261M					78 (7) 391x	>100 >1429x	
**AG110^res^** A221K,R E291G					3.7 (0.7) 19x	>100 >1429x	
**BBP^res^** I261M					5 (1) 25x	>100 >1429x	
**TO502-2403^res^** F224P	>100 11x	79 (5) 1580x	>100 63x	60 (6) 200x	>100 >500x	>100 >1429x	14 (2)
**TO505-6180^res^** F224Y N264D	>100 11x	45 (3) 900x	25 (4) 16x	>100 >333x	32 (6) 160x	>100 >1429x	2.5 ± 0.4
**Wild type (NADL)**					0.20 (0.06)	0.07 (0.02)	

Effective concentration 50% values (EC_50_) are average values for 3 independent experiments; Values between brackets represent SD; n.d.: not determined; Values followed by x indicate fold resistance.

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
