# Peer review of "Quinolinecarboxamides Inhibit the Replication of the Bovine Viral Diarrhea Virus by Targeting a Hot Spot for the Inhibition of Pestivirus Replication in the RNA-Dependent RNA Polymerase"

_molecules, 2020, doi:10.3390/molecules25061283_

Round 1

Reviewer 1 Report

This is an interesting work, data are well presented to support the conclusion. But this manuscript is far not complete, there are many writing errors and necessary information missing, reflecting the authors’ carelessness.

It is better to give a brief introduction of BVDV and its threaten in the very beginning of abstract.

Introduction line 46-47 can be combined in adjacent paragraph because too short to be an independent paragraph.

There are a number of format errors to be corrected in the manuscript. For example, line 69, a big space; line 71 (iii) not italic; line 136, 2.5 sub-title; and many other.

Line 248, the authors identified two quinolinecarboxamides from a large scale screening, but these is no information in the manuscript about what library they used and the relative methods. And, does the library contain many quinolinecarboxamide analogues?

Fig 2, what does UTC stand for? Legend needs to be rewritten because confused sentence structure.

3.3 line 333 334, which is table 11?

Fig 5 legend, where are descriptions of panel C and D?

Fig 6 legend, where is description of panel B?

Author Response

We responded as follows to the comments:

Reviewer #1:

  1. This is an interesting work, data are well presented to support the conclusion. But this manuscript is far not complete, there are many writing errors and necessary information missing, reflecting the authors’ carelessness.

We now have thoroughly proofread the manuscript and amended errors and formatting issues.

  1. It is better to give a brief introduction of BVDV and its threaten in the very beginning of abstract.

We now have modified the abstract according to the reviewer’s instructions.

  1. Introduction line 46-47 can be combined in adjacent paragraph because too short to be an independent paragraph.

We now merged line 46-47 with the adjacent paragraph.

  1. There are a number of format errors to be corrected in the manuscript. For example, line 69, a big space; line 71 (iii) not italic; line 136, 2.5 sub-title; and many other.

Thank you very much for thoroughly proofreading our manuscript. We now amended the mistakes identified and as mentioned in our answer on your first remark we proofread the manuscript.

  1. Line 248, the authors identified two quinolinecarboxamides from a large scale screening, but these is no information in the manuscript about what library they used and the relative methods. And, does the library contain many quinolinecarboxamide analogues?

We now included a reference and a brief description of the library and methods used to identify this compound.

  1. Fig 2, what does UTC stand for? Legend needs to be rewritten because confused sentence structure.

            UTC stands for untreated control. We now mention this in the legend to Fig. 2.

  1. 3 line 333 334, which is table 11?

We now refer to the correct table 1.

  1. Fig 5 legend, where are descriptions of panel C and D?

We now added a description of panel C and D to the caption of Fig 5.

  1. Fig 6 legend, where is description of panel B?

We now added a description of panel B to the caption of Fig 6.

We like to thank the reviewer for her/his time and critical review of our manuscript.

Best regards,

Jan Paeshuyse

Reviewer 2 Report

Musiu and colleagues identified two inhibitors of BVDV replication.  These quinolinecarboxamide analogues were found to inhibit the virus in concentrations lower than 1 micro M, including reduction of CPE, infectious virus and extra and intracellular viral RNA. The authors perform Quantitative structure-activity relationships analyses in silico to understand the bioactivity of these compounds.  They also find that these drugs do not inhibit purified viral polymerase activity in vitro but do inhibit activity of isolated replication complexes. They also perform in vitro docking of the drugs with the polymerase to reveal possible interactions. 

Major Comments:

1. The authors mentioned that they identified these compounds during a large-scale CPE-based screening. However, no details are given regarding that screen, was this previously described in another paper? If not, the authors sohuld give details about how the screen was done, how many compounds were screened and a table with all of the results.

2. In figure 2, how do the authors explain that 0.1 micro M inhibit intra and extracellular RNA completely but did not inhibit infectious virus? These data appears to be contradictory.

3. I was not fully convinced that the mutations in NS5B make the virus resistant to the drugs. The resistant viruses which were passaged in the presence of the drugs likely have other mutations in other proteins as well. Other mutations should be mentioned and how the authors selected the 2 NS5B mutations. Additionally, to show the mutations in NS5B are important, a mutant virus made through reverse genetics should be tested that contain only these 2 mutations. Although the authors show that the drugs work in vitro by isolating the RC; in this assay the drugs could also be interacting with host factors and indirectly impacting replication.

4. In figure 2, the authors describe their experiments as % of untreated control. A better control would be vehicle control, did the authors use a vehicle in their "UTC"?

5. How pure was the extraction of the replication complexes? Did the authors have controls to see what other cellular factors were purified with the RCs (eg. ribosomes)?

Other comments:

The paper has a number of inconsistencies that impede comprehension and should be fixed: Many of the paragraphs are composed by a single sentence and appear to be disconected from the rest of the text. I could not find the supplemental tables 1 and 2, were they included? In figure 4`s legend, two of the graphs are described as "filled squares". Figure 5`s legend is missing letter`s C and D. In Figure 5A the concentration increases towards the left, whereas in 5C it increases towards the right, this should be standardized.  Figure 6`s legend is missing letter B. In several parts of the text "table 11" is mentioned. I believe this is a typo (for example, line 332). In line 136, the numbering is off, after "2.4" comes "2.5.3".

2. Figure 1 is not mentioned in the text.

3. Figure 1 and table 1 are in the middle of the methods section. It would make more sense for them to be in the results section, where they are mentioned.

4. In line 50-51, the authors should mention what strains they are referring to. 

Author Response

We responded as follows to the comments of Reviewer #2:

Musiu and colleagues identified two inhibitors of BVDV replication.  These quinolinecarboxamide analogues were found to inhibit the virus in concentrations lower than 1 micro M, including reduction of CPE, infectious virus and extra and intracellular viral RNA. The authors perform Quantitative structure-activity relationships analyses in silico to understand the bioactivity of these compounds.  They also find that these drugs do not inhibit purified viral polymerase activity in vitro but do inhibit activity of isolated replication complexes. They also perform in vitro docking of the drugs with the polymerase to reveal possible interactions. 

Major Comments:

  1. The authors mentioned that they identified these compounds during a large-scale CPE-based screening. However, no details are given regarding that screen, was this previously described in another paper? If not, the authors sohuld give details about how the screen was done, how many compounds were screened and a table with all of the results.

We now included a reference to the library and methods used to identify this compound.

  1. In figure 2, how do the authors explain that 0.1 micro M inhibit intra and extracellular RNA completely but did not inhibit infectious virus? These data appears to be contradictory.

Not necessarily, extracellular RNA is determined using RT-qPCR that has a lower limit of detection. Furthermore, viral RNA has to be extracted from cell culture supernatant and the recovery rate of such an extraction is close to 100%. Determining the infectious virus yield from a treated culture requires incubation of infectious viruses in cell culture. Which implies that low concentrations of infectious virus can amplify themselves. Thus, the difference observed between infectious virus titers and concentration of intra- or extracellular viral RNA is in our opinion more related to the detection method used.

  1. I was not fully convinced that the mutations in NS5B make the virus resistant to the drugs. The resistant viruses which were passaged in the presence of the drugs likely have other mutations in other proteins as well. Other mutations should be mentioned and how the authors selected the 2 NS5B mutations. Additionally, to show the mutations in NS5B are important, a mutant virus made through reverse genetics should be tested that contain only these 2 mutations. Although the authors show that the drugs work in vitro by isolating the RC; in this assay the drugs could also be interacting with host factors and indirectly impacting replication.

Indeed, introducing the mutation identified in a WT background would prove that this point mutation is responsible for the resistant phenotype observed. However, due to technical and logistical reasons (key authors moved to other labs) we do not have these data. We do have indirect evidence using a reverse engineered virus in which a single point mutation was introduced, i.e. F224S mutation that confers resistance to BPIP. We obtained this virus through a collaboration with Prof R Donis in 2004. Since then Prof R Donis moved to other scientific fields and the expertise/technology is no longer available. Reverse engineered BPIPres F224S is >1429 times less susceptible to inhibition by TO505-6180 and 391-times less susceptible to inhibition by TO502-2403. This cross resistance profile shows that a single mutation at position F224 suffice to confer resistance to TO505-6180 and TP502-2403.

  1. In figure 2, the authors describe their experiments as % of untreated control. A better control would be vehicle control, did the authors use a vehicle in their "UTC"?

To the UTC we always add vehicle or buffer without compound similar to the treated conditions.

  1. How pure was the extraction of the replication complexes? Did the authors have controls to see what other cellular factors were purified with the RCs (eg. ribosomes)?

No, we did not do a proteomic analysis of the replicase complex extracts. This is beyond the scope of this publication.

Other comments:

The paper has a number of inconsistencies that impede comprehension and should be fixed:

  1. Many of the paragraphs are composed by a single sentence and appear to be disconected from the rest of the text.

We proofread the manuscript and amended our paragraph use according to the reviewer’s instructions.

  1. I could not find the supplemental tables 1 and 2, were they included?

Thank you for noticing this. We now have included the Supplemental tables.

  1. In figure 4`s legend, two of the graphs are described as "filled squares".

We now added filled diamonds so all graphs in figure 4 are also correctly referred to in the caption of figure 4.

  1. Figure 5`s legend is missing letter`s C and D. In Figure 5A the concentration increases towards the left, whereas in 5C it increases towards the right, this should be standardized.

We now included letters C and D with additional text to explain the panels to the caption of figure 5. In addition we now have standardized how concentration gradients are depicted in Fig. 5.

  1. Figure 6`s legend is missing letter B.

We now added letter B to the legend of figure 6.

  1. In several parts of the text "table 11" is mentioned. I believe this is a typo (for example, line 332).

Indeed, table 11 should have been table 1. We now corrected this throughout the manuscript.

  1. In line 136, the numbering is off, after "2.4" comes "2.5.3".

We now corrected the numbering.

  1. Figure 1 is not mentioned in the text.

We now refer to Figure 1 at the beginning of the material and methods section and at the beginning of the results section.

  1. Figure 1 and table 1 are in the middle of the methods section. It would make more sense for them to be in the results section, where they are mentioned.

In the original manuscript file submitted for review all figures were at the end of the manuscript. I suppose that this happened during formatting of the submitted file in the journal’s layout.

  1. In line 50-51, the authors should mention what strains they are referring to.

We now mention BVDV-1 and -2 strains.

We like to thank the reviewer for her/his time and critical review of our manuscript.

Best regards,

Jan Paeshuyse